# Statistical Analysis of Grain-Scale Effects of Twinning Deformation for Magnesium Alloys under Cyclic Strain Loading

**DOI:** 10.3390/ma13112454

**Published:** 2020-05-28

**Authors:** Damin Lu, Shuai Wang, Yongting Lan, Keshi Zhang, Wujun Li, Qixi Li

**Affiliations:** 1School of Vocational and Technical Education, Guangxi University of Science and Technology, Liuzhou 545006, China; luda.min@163.com (D.L.); liwujun6601@126.com (W.L.); liqixi1019@163.com (Q.L.); 2Key Lab of Disaster Prevent and Structural Safety, Guangxi Key Lab Disaster Prevent and Engineering Safety, College of Civil Engineering and Architecture, Guangxi University, Nanning 530004, China; zhangks@gxu.edu.cn; 3School of Mechanical and Transportation Engineering, Guangxi University of Science and Technology, Liuzhou 545006, China; lyt_456@126.com

**Keywords:** twinning deformation, grain-scale, magnesium alloy, cyclic loading, statistical analysis

## Abstract

To reveal the relationship between grain size and twinning deformation of magnesium alloys under cyclic strain, this study carried out a group of strain-controlled low-cycle fatigue experiments and statistical analysis of microstructures. Experimental results show that the shape of the hysteresis loop exhibits significant asymmetry at different strain amplitudes, and the accumulation of residual twins plays an important role in subsequent cyclic deformation. For the different strain amplitudes, the statistical distribution of the grain size of magnesium alloy approximately follows the Weibull probability function distribution, while the statistical distribution of twin thickness is closer to that of Gaussian probability function. The twin nucleation number (TNN) increases with the increase of grain size, but there is no obvious function relationship between twin thickness and grain size. Twin volume fraction (TVF) increases with the increase of grain size, which is mainly due to the increase of TNN. This work can provide experimental evidence for a more accurate description of the twinning deformation mechanism.

## 1. Introduction

The component of Magnesium alloy is often subject to cyclic loading in service. Many experiments show that the evolution of the stress-strain curve, yield strength and hardening rate of magnesium alloy varies significantly with different loading paths [1,2,3,4]. Especially, magnesium alloy presents strong Bauschinger-effect and remarkable asymmetric plastic flow behavior during strain-controlled cyclic tension-compression tests [5,6]. In the majority of symmetric strain-controlled fatigue tests of Magnesium alloy [7,8], dislocation slip and twinning-detwinning deformation are two important patterns of plastic deformation. Dislocation slip acts as the main mechanism of plastic deformation when the applied strain amplitude is relatively small, while twinning-detwinning deformation plays a predominant role during cyclic deformation at high strain amplitudes. However, experiments [9] show that there are considerable discrepancies of intrinsic properties between dislocation slip and twinning deformation. Dislocation slip mainly appears as a thermal activation process, which is the function of temperature and strain rate. However, twinning deformation lacks sensitivity to temperature and strain rate effects. Although the high critical resolved shear stress (CRSS) is the main factor for crystal twinning system generating twinning deformation [10,11], the majority of twin nucleation originates from grain boundaries, dislocation stacking, defects, slip bands, twin interfaces and cracks [12,13].

The alternate occurrence of twinning and detwinning deformation is a very important deformation mechanism during low-cycle fatigue tests of magnesium alloy. By the in-situ neutron diffraction method [14], it can be found that twinning and detwinning activity involves in the deformation process during cyclic loading of magnesium alloy. In literature [15], symmetric strain-controlled cyclic tests on extruded ZK60(Mg-Zn6%-Zr0.3% alloy) found that according to the occurrence degree of twinning deformation or detwinning deformation, there are three deformation mechanisms under different strain amplitudes: dislocation slip, complete twinning/partial detwinning and complete twinning/detwinning. By using transmission electron microscopy (TEM) and electron backscattering diffraction (EBSD) to observe the characteristics of microstructure evolution of magnesium alloy under cyclic loading [16,17], it can be seen that the cyclic hardening, which is caused by twinning/detwinning deformation and dislocation slip, increases the site of twinning nucleation, but inhibits twin growth and twin shrinkage. Experiments [18] found that twinning deformation (i.e., TVF) increases with the number of loading cycles increases, but twinning deformation will reach saturate state after the number of loading cycles rises to a certain value. As the strain amplitude increases, the grains with a little TVF will also generate more twins and form residual twin boundaries. This characteristic dominates the low-cycle fatigue behavior of magnesium alloy. The macroscopic stress-strain response of the material is intrinsically related to the micro-structures of the material during cyclic loading. The stability of twin boundaries is influenced by TVF, twin morphology, and cyclic hardening [14].

Although the importance of twinning and detwinning deformation mechanism during the cyclic deformation of magnesium alloys has been widely recognized in many previous studies, both the inhomogeneous deformation of twinning and detwinning and the inhomogeneous distribution of residual twins in polycrystalline are still not systematically studied. In particular, the statistical relationship between twinning deformation and microstructure evolution remains ambiguous. The twin formation process can be divided into the formation and the growth of the fine twin region, i.e., twin nucleation and growth [19]. The pattern of twin nucleation is that local dislocations are arranged and decomposed into single or multiple stacking faults which then form twin nuclei. The process of twin growth is that the twin nucleation on the twin plane is surrounded by partial twin dislocation of the matrix, which makes the twins on the twin plane expand rapidly. Studies at the atomic scale successfully revealed the process of twin growth by the reaction between the twin boundary and the lattice dislocation [20]. However, twin nucleation is still a subject that has not been deeply studied. A series of studies [21,22] showed that the crystal orientation is defined by the Schmid factor that determines which twin variants would dominate. The Schmid factor of the twinning system has a more significant effect on twin nucleation than twin thickening in magnesium alloy [23]. Twin nucleation and growth are not affected by dislocation slip during twinning deformation [24]. Researchers carried out studies on the grain size effect of metal crystals such as Zr [25], Ti and AZ31(Mg-Al3%-Zn0.9% alloy) [26,27] and pointed out that the influence of grain size effect on the nucleation pattern and growth mechanism of each twin is greatly different during deformation. By studying the size effect of twinning deformation of micro-column of pure magnesium crystals [28], it is found that no twins can be seen in the micro-columns with a diameter of 2–10 nm, but twinning deformation occurred in the samples with a diameter of about 200 nm, and the defect nucleation plays an important role in twinning deformation. The above studies reflected that twinning deformation is closely linked with grain orientation and grain size. Although current studies have reported that twins are formed in large grains and more likely to occur in low-angel boundary grains or preferred orientation grains, but not all large grains, low-angle boundary grains or grains of the same size but with a preferred orientation generate twinning deformation [29,30]. To reveal the relationship between twinning deformation and grain size under cyclic loading, the symmetric tension-compression experiments with different strain amplitudes were carried out and statistical analysis of the microstructural information such as grain area, twin thickness and the TNN after cyclic deformation was conducted in this paper.

## 2. Material and Experimental Procedure

### 2.1. Experimental Material and Specimen

The extruded AZ31 magnesium alloy bars with a diameter of 25 mm were commercially acquired, the chemical composition of the material was listed in Table 1. The as-received materials were annealed for 1h at 400 °C in a vacuum furnace (Wuhan hankou electric furnace Co. LTD, Wuhan, China) to yield a microstructure of twin-free. All bars were machined into dog-bone-shaped specimens with a gauge length of 50 mm and a diameter of 16.5 mm within the gauge section. The loading axis of the specimen was aligned with the direction of extrusion, the detail geometry and dimensions of the testing specimen were shown in Figure 1. To eliminate the influence of surface roughness on fatigue performance, the outer surface of the gauge section of each testing specimen was ground mechanically using SiC sandpapers (Sui Sun Co., LTD, Hong Kong, China) with grit number from 250 up to 2000 before fatigue tests step by step.

### 2.2. Experimental Procedure

All low-cycle fatigue tests were conducted under a sinusoidal-waveform-strain-controlled mode with a frequency of 0.5 Hz at room temperature by using an MTS809 axial-torsional servo-hydraulic machine (MTS Systems Corporation, Eden Prairie, MN, USA). The testing system has a capacity of ±250 kN for axial force. For all strain-controlled fatigue experiments, an extensometer (MTS Systems Corporation, Eden Prairie, MN, USA) with a gauge length of 25 mm and a range of ±10% was mounted on the outer surface of the gauge section of specimens to measure the axial strain during cyclic loading. And about 200 data points of strain and force are recorded during every cycle. In this study, the total strain amplitudes applied were 0.3%, 0.5%, 0.8%, 1.2%, 1.5%, and 2%, respectively, and at least two specimens were tested at each strain amplitude level. Fatigue failure for each specimen was considered to occur when the drop of tensile peak load reached 10% of the half-life tensile peak load, the average fatigue life under different strain amplitudes was shown in Table 2.

To make statistical analysis of microstructure information about the samples after cyclic failure, the cylindrical samples with a thickness of about 10 mm were cut from the gauge section of test specimens near the fracture surface, the observing plane was perpendicular to the extrusion direction. The samples for cross-sectional optical observation were first ground mechanically by using SiC sandpapers with grit number from 250–2500 step by step, and further polished with a diamond paste of 5 μm, 3.5 μm, 2.5 μm, 1.5 μm, and 1 μm. The surface oxide layer of the samples was eliminated with a dilute nitric acid solution (1% concentration), then etched with a picric acid solution (Picric acid 5.5 g, acetic acid 10 mL, and 70 mL ethanol with 95% concentration) for 15–25 s. The microstructures of etched samples were captured by a metallographic microscope (Leica Instruments Co., LTD., Heidelberg, Germany), photographed and preserved. Finally, we obtain metallographic microstructure pictures corresponding to each selected strain amplitude as shown in Figure 3.

## 3. Experiment Results 

### 3.1. Cyclic Deformation under Tension-Compression

Figure 2 presents typical stress-strain hysteresis loops curves obtained from fully reversed strain-controlled fatigue experiments at six strain amplitudes (i.e., half of total strain range) of 0.3%, 0.5%, 0.8%, 1.2%, 1.5%, and 2%, respectively. To clearly show the evolution of hysteresis loop shape with different loading cycles, for every applied strain amplitude, the stress-strain hysteresis loops of the first two cycles, half-life cycle, and 90% cycle of fatigue life are plotted in a single graph, respectively. It is clear that the shape of the hysteresis loop mainly depends on both the controlled strain amplitudes and the number of loading cycles. Additionally, the different levels of cyclic hardening are observed under cyclic tension-compression loading for certain strain amplitude range.

For different strain amplitudes, it is worth noting that when the peak stress of the first cycle is unloaded, an elastoplastic transition occurs. The flow stress maintains relatively steady with further plastic deformation which forms approximately zero-work hardening plateau during compression. This phenomenon is more obvious with the increase of strain amplitude. When unloading from the compressive peak stress and reverse loading, a significant Bauschinger effect is observed. In other words, the linear elastic part is fairly small, and the material exhibits a pseudo-elastic behavior beyond the reverse yield point, which can be caused by the detwinning deformation mechanism. When the strain reaches a certain value, the stress-strain hysteresis loop appears inflection point and the hardening rate rapidly increases due to the exhaustion of the detwinning mechanism beyond this inflection point. During loading of the second cycle, twinning and detwinning deformation occur alternately, which results in the asymmetric sigmoidal-shaped loop and obvious asymmetry hardening behavior between the compression half-cycle and the tension half-cycle. Compared with the first cycle at high strain amplitudes of 0.8%, 1.2%, 1.5%, and 2.0%, the most obvious change of second hysteresis loop is that elastic-plastic transition becomes slower and non-zero hardening behavior occurs during unloading in the compression half cycle, the plateau also becomes shorter than that in the first cycle. Those phenomena show that twins generating in grains during the compression process of the first cycle do not recover completely during reversal reloading and the residual twins are formed. The residual twins have an important influence on the cyclic deformation behavior in the second cycle. More obvious effect can be observed as strain amplitude increases due to stronger twinning–detwinning deformation characteristic at high strain amplitudes. With the increase of the number of cycles, both the maximum tensile peak stress and the maximum compressive peak stress increase to varying degrees, showing the overall cyclic hardening behavior. The cyclic hardening is mainly due to the increase of twin boundary area and twin fracture and dislocation increase in density.

In summary, a large amount of twin deformation occurs in the compression stage of cyclic loading, then twin deformation recovery occurs in the subsequent unloading stage. Furthermore, although the twinning deformation occurs during unloading or reverse loading, not all twins are fully recovered, and some twins remain in the grains, which are called residual twins. The accumulation of residual twins in grains has an important influence on subsequent cyclic deformation.

### 3.2. Microstructure Observation

Figure 3 shows the metallographic microstructures of specimens after cyclic failure in which obvious twinning deformation can be seen under six specific strain amplitudes, and the number of twins increases with the increasing strain amplitude. In terms of the number of strip-shaped twins in single grain, the number of twins is relatively small in the grains at the strain amplitude of 0.3%, and a considerable amount of grains don’t generate twins. As strain amplitude increases, the number of twins in single grain also increases, and the number of grains that generate twinning deformation increases as well. In terms of the morphology of the twins, the twins of grains at low strain amplitudes are mainly sparse parallel twins (A, B), while parallel twins become denser as strain amplitude increases (C, D). With the further increase of strain amplitude, intersecting twins (E) gradually appear which indicates that the presence of more than two kinds of twin variants leads to severe fragmentation of grains (F). Regarding the thickness of strip-shaped twins, the width of strip-shaped twins at different strain amplitudes shows little differences. But at the same strain amplitude, there is a large difference among twins of different-size grains.

The above analysis shows that twinning deformation becomes more and more complicated with the increase of strain amplitude, and the twinning deformation in different grains also has an obvious difference. Therefore, it is necessary to further reveal the relationship between twinning deformation and grain size under cyclic loading by the statistical method. 

## 4. Result Analysis and Discussion

### 4.1. Statistical Method of Microstructures Parameters

The microstructure information of different regions on each sample was randomly selected, photographed and preserved to obtain a large number of microstructure images. Each grain region in the metallographic picture was marked with a closed multi-segment line (as shown by the red box in Figure 4) and was collected as a grain area. Each band-shaped layer (twin) observed in every grain was counted as one twin nucleation (signified by T_1_, T_2_, T_3_, T_4_ indicating that there were four twin nuclei in this specific grain as shown in Figure 4). The width of the twin layer was taken as the thickness of the twin nucleation (signified by W_1_, W_2_, W_3_, W_4_ in Figure 4, which stands for the thickness of twin T_1_, T_2_, T_3_, T_4_ respectively). The formation of twins means that twinning deformation participates in plastic deformation behavior, so twinning behavior can be well explained by statistical analysis of twins. Twinning deformation includes two processes, namely twin nucleation and twin growth (thickening). This paper defines one twin band as one twin nucleation. The statistics of microstructure, such as the number of twin nuclei and the value of twin thickness, will help us further understand the relationship between twinning deformation and grain size and other microstructure information.

The microstructure of the samples at different strain amplitudes was statistically analyzed, and the microstructure data such as grain area, the TNN and twin thickness were obtained. These collected data were aggregated and further processed in detail, the relationship between different data was represented by the form of histograms and scatter diagrams.

### 4.2. Statistical Distribution of Grain Size

Figure 5 presents histograms of the distribution about grain area size in magnesium alloy corresponding to the strain amplitude of 0.3%, 0.5%, 0.8%, 1.2%, 1.5% and 2.0%, respectively. It can be seen in the histograms that the grains with a range from 1500 μm^2^ to 2500 μm^2^ area occupy the largest proportion in the whole grain size distribution; the grains whose area is smaller than 1500 μm^2^ or bigger than 2500 μm^2^ take smaller proportion. The quantitative distribution of grain size approximately follows the Weibull probability distribution. Based on the quantitative distribution characteristics of grain area, we adopt Weibull probability function to fit the distribution of grain area and obtain the fitting curves of grain area probability distribution that correspond to specific strain amplitude, as shown in Figure 5.

Table 3 shows the average grain size corresponding to different strain amplitudes in which we can see directly that the average grain size is mainly distributed between 2000 μm^2^ and 2200 μm^2^. Here, the average grain size is defined as the sum of all the individual grain areas divided by the total number of grains within the range of the captured image, and only the grains with a complete area are considered. The difference of average grain size at different strain amplitudes is relatively small, which means that the cyclic loading under different strain amplitudes doesn’t lead to a large change of total grain size, but the twin fragmentation that is observed in the magnesium alloy metallographic microstructure at relatively high strain amplitudes may help refine the grains during further cyclic loading. 

### 4.3. Statistical Distribution of Twin Thickness Number

The histograms (a–f) as shown in Figure 6 present the distribution of twin thickness number and its approximately Gaussian fitting curve. In the case of different strain amplitudes, the twin thickness is mostly distributed in a certain range of thickness which accounts for the highest proportion. The overall distribution is consistent with the approximate Gaussian probability distribution. Regarding the distribution of twin thickness value, twin thickness value is mainly distributed between 1.7 μm and 2.4 μm and there is no significant change at different strain amplitudes. It can be attributed to the accumulation of residual twins during the cyclic deformation process. Although the strain amplitude is different, the specimens at low strain amplitudes go through more cycles which will result in the accumulation of more residual twins.

### 4.4. Relationship between TNN and Grain Size

As shown in Figure 7, Diagrams (a–f) display the relationship between grain area and the TNN at the strain amplitudes of 0.3%, 0.5%, 0.8%, 1.2%, 1.5% and 2.0% respectively. The scatters in every diagram represent the TNN corresponding to different grain area. The relationship between the TNN and grain size at different strain amplitudes shows similar growth trends. If we further fit the relationship between the size and the distribution of TNN linearly. Although the slope of those lines does not increase strictly as the strain amplitudes increase, but they show an increasing trend on the whole, except that the slope gets smaller at strain amplitudes of 0.5% and 1.5%, relatively. This phenomenon reflects directly that the growth rate of twin numbers increase more rapidly in the case of large strain amplitudes within the same grain size range. However, it can be seen from the statistical results that the grain area size corresponding to certain amount of TNN is limited in a certain range, or in other words, grains of the same area size can only generate a finite TNN. Based on this point, twin nucleation should also be influenced by other factors such as grain orientation, grain boundary and so on.

Table 4 lists data of twin nucleation, such as the maximum TNN, the average TNN and the minimum TNN at different strain amplitudes. Based on the TNN inside the grains, we can infer that the TNN is relatively small at low strain amplitudes. For example, the TNN ranges from 1 to 8 at the strain amplitude of 0.3%, but that ranges from 3 to 15 at the strain amplitude of 2.0%, which demonstrates that the TNN increases with the increase of strain amplitude as a whole.

As shown in Figure 8, Diagrams (a–f) show the variation of the TNN as a function of unit grain area (in this paper 1 unit area stand for 1000 μm^2^). The TNN within per unit grain area at different strain amplitudes share basically the same trend, which mainly reflects that the TNN per unit area decreases with the increase of grain area. This phenomenon indirectly reflects that the TNN per unit volume in material decreases with the increase of grain size and this tendency tends to the relation of the fractional exponential function.

Based on the above analysis, the TNN can increase with the increase of strain amplitude during the cyclic loading. At different strain amplitudes, the TNN also increases with the increase of grain size and the TNN of a certain grain area changes within a limited range. However, the TNN per unit grain area decreases exponentially with the increase of grain area within a single grain.

### 4.5. Relationship between Twin Thickness and Grain Size

Diagrams (a–f) as in Figure 9 present the relationship between grain size and the average twin thickness in the grain. Each scatter point in the diagram represents the average value of all twin thickness in a single grain. It can be seen by comparison that the distribution of the average value of twin thickness is not regular, and there is no obvious functional correlation between grain area size and the average value of twin thickness. The distribution range of all twin thicknesses under different strain amplitudes is relatively similar, which means that after undergoing different cyclic deformations during cyclic loading, the accumulation of residual twins reaches a relatively stable thickness value. The polyline in the figure shows the variation trend of the average value of all twin thickness within each interval of 500 μm^2^ range in all grains. The average value of twin thickness shows wave-like fluctuation instead of monotonic variation. It can be seen that the process of twin thickness growth is less affected by the grain size and no significant size-effect was shown.

As shown in Table 5, the average twin thickness of all grains remains around 2 μm at different amplitudes, and the upper and lower bounds of twin thickness basically remain within a similar numerical range, which reflects that the accumulation of residual twin tends to approach the same target value during cyclic deformation of the material at different strain amplitudes. 

### 4.6. Relationship between TVF and Grain Size

The TVF is defined as the ratio of the total area of all strip-shaped twins to the area of the grain in a single grain. In Figure 10, Diagrams (a–f) show the relationship between TVF and grain size at applied strain amplitudes. The multi-segment lines in the diagrams represent average volume fraction of all grains within each interval of 500 μm^2^ grain area range. As can be seen from the figure, the TVF in grains generally increases with the increase of grain area at different strain amplitudes. Regarding the variation of the average value of TVF, the multi-segment line shows a flexuous upward trend, which indicates that the increase of grain size is conducive to the generation of twin deformation. If we further fit the relationship between the grain size and TVF linearly according to the method shown in Figure 7, the slopes of these lines show the similar trend, which reflects directly that the ratio of TVF is higher within the same grain size range at high strain amplitudes.

Since twin thickening is not sensitive to the grain size according to the results shown Figure 9, the increase of TVF is mainly reflected by the increase of TNN. By comparing the distribution of TVF at different strain amplitudes, it can be seen that TVF increases with the increase of strain amplitude. For example, the maximum TVF can only reach about 25% at low strain amplitude of 0.3%, while the maximum TVF could reach about 50% at high strain amplitude of 2%. In conclusion, the increase of grain size is beneficial to twin nucleation. Grain size has an obvious effect on twin nucleation but has little effect on twin growth

## 5. Conclusions

This study carried out tests of symmetric cyclic tension-compression loading under strain control at six different strain amplitudes on AZ31 magnesium alloy, made a statistical analysis of microstructures such as grain size, twinning deformation and so on, analyzed the relationship between macroscopic cyclic hysteresis loop curve of magnesium alloy and twinning deformation, made further statistical investigation into the relationship between grain size and the distribution of twins and obtained the following conclusions:The cyclic hysteresis loop curves with different strain amplitudes show significant asymmetry. Twinning deformation has a severe influence on the evolution of macroscopic hardening rate. The accumulation of residual twins per cycle plays an important role in subsequent cyclic deformation.Under different strain amplitudes, the statistical distribution of grain size of magnesium alloy after cyclic deformation approximately conforms to Weibull probability function distribution, but the statistical distribution of twin thickness is closer to the Gaussian distribution characteristic.The TNN always increases with the increase of grain size at different strain amplitudes, and the TNN of a certain grain area varies within a finite range. However, the number of nucleation per grain area decreases with the increase of grain area.The distribution range of twin thickness at different strain amplitudes is quite close, which indicates that after undergoing different cyclic deformations during cyclic loading, the accumulation of residual twin reaches a relatively stable thickness value.The TVF increases with the increase of grain area at different strain amplitudes. Under the condition that there is little difference in twin thickness, the increase of TVF is mainly due to the increase of the TNN.

## Figures and Tables

**Figure 1 materials-13-02454-f001:**
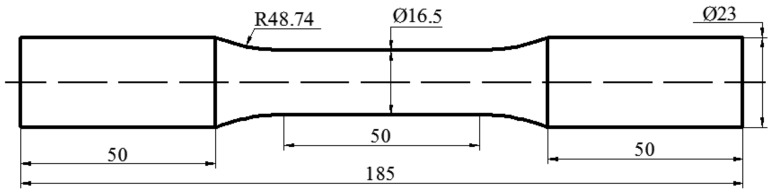
Shape and dimensions of the sample (unit:mm).

**Figure 2 materials-13-02454-f002:**
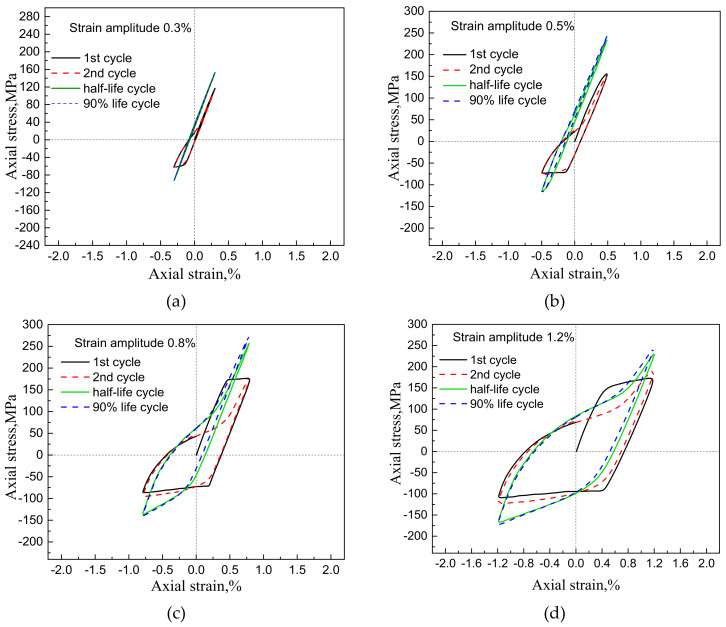
Cyclic hysteresis loops corresponding to different cyclic numbers at the selected strain amplitudes: (**a**) 0.3%; (**b**) 0.5%; (**c**) 0.8%;(**d**) 1.2%;(**e**) 1.5%; (**f**) 2.0%.

**Figure 3 materials-13-02454-f003:**
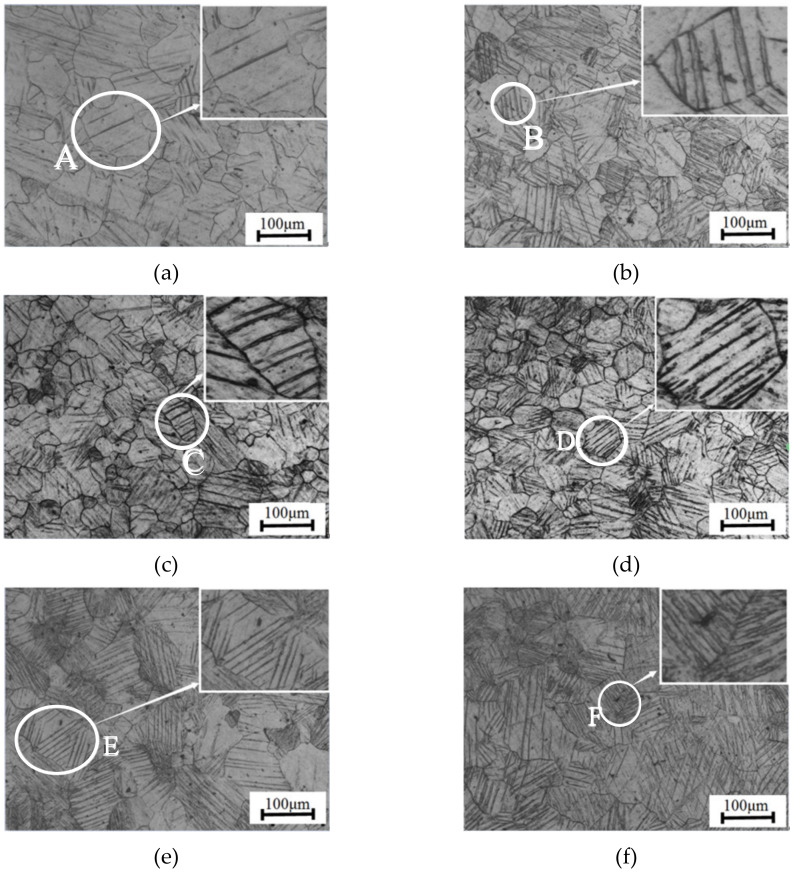
Metallographic microstructure at different strain amplitudes: (**a**) 0.3%; (**b**) 0.5%; (**c**) 0.8%; (**d**) 1.2%; (**e**) 1.5%; (**f**) 2.0%.

**Figure 4 materials-13-02454-f004:**
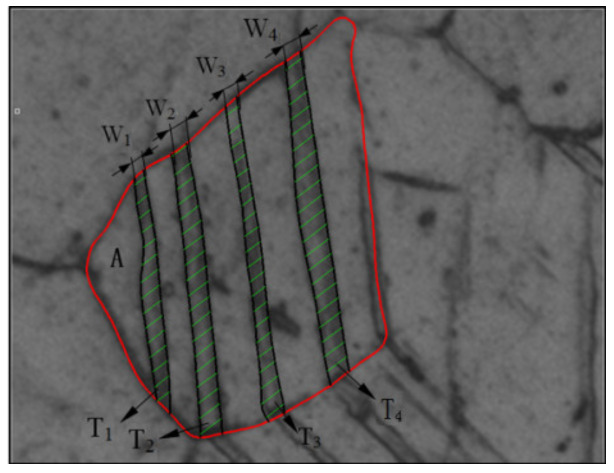
Schematic illustration of microstructural statistical variables.

**Figure 5 materials-13-02454-f005:**
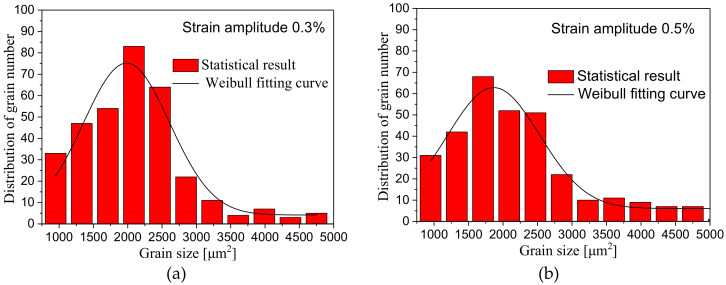
Distribution diagram of grain area size corresponding to different strain amplitudes: (**a**) 0.3%; (**b**)0.5%; (**c**) 0.8%; (**d**) 1.2%; (**e**) 1.5%; (**f**) 2.0%.

**Figure 6 materials-13-02454-f006:**
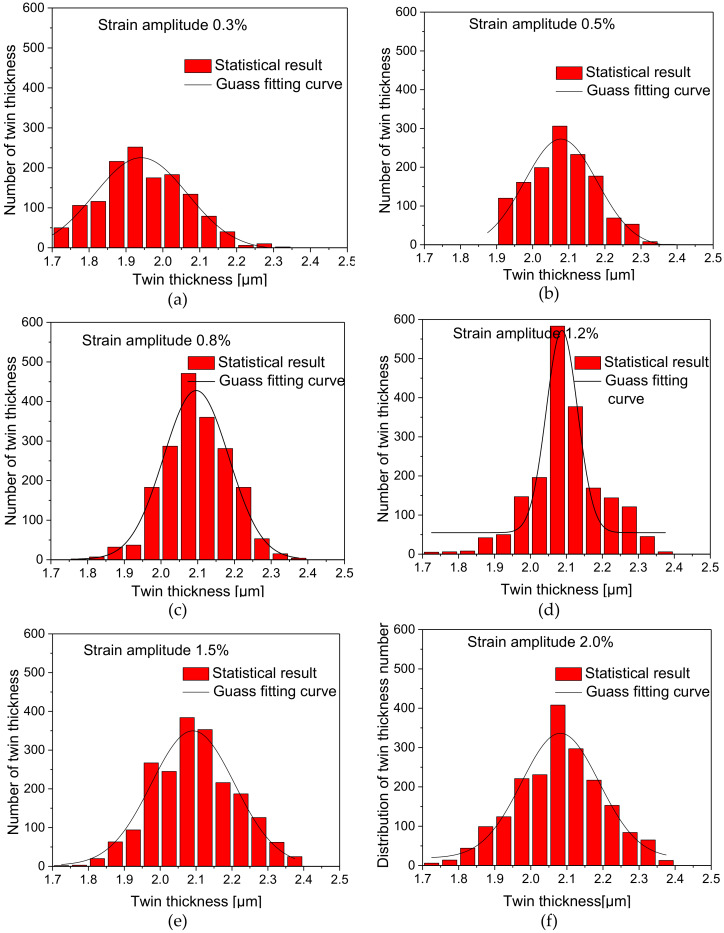
Distribution diagram of twin thickness corresponding to different strain amplitudes: (**a**) 0.3%; (**b**) 0.5%; (**c**) 0.8%; (**d**) 1.2%; (**e**) 1.5%; (**f**) 2.0%.

**Figure 7 materials-13-02454-f007:**
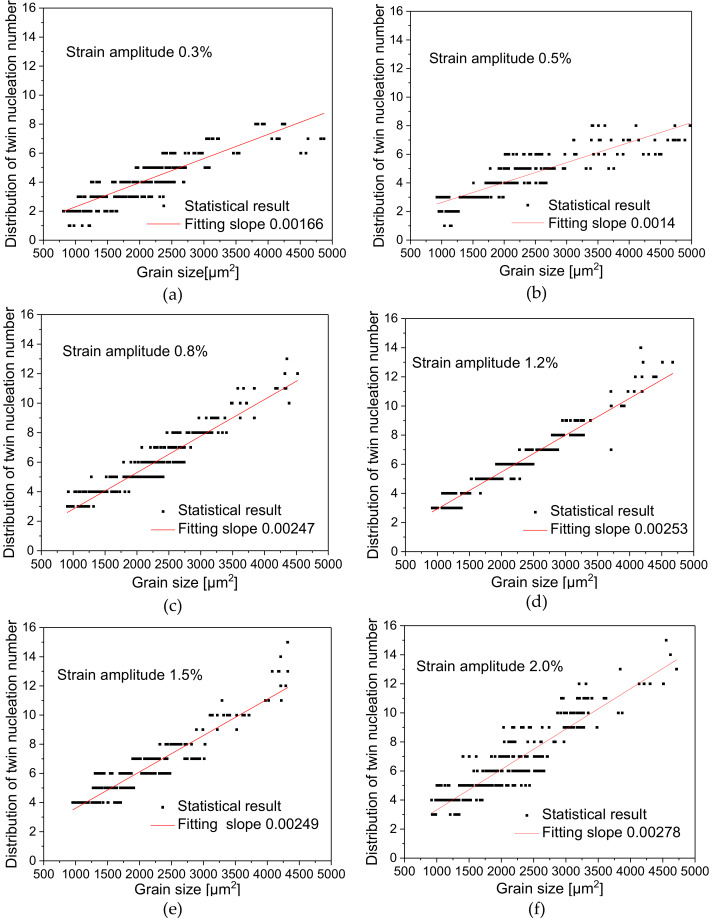
Distribution diagram of the relationship between grain area size and TNN at different strain amplitudes:(**a**) 0.3%; (**b**) 0.5%; (**c**) 0.8%; (**d**) 1.2%; (**e**) 1.5%; (**f**) 2.0%.

**Figure 8 materials-13-02454-f008:**
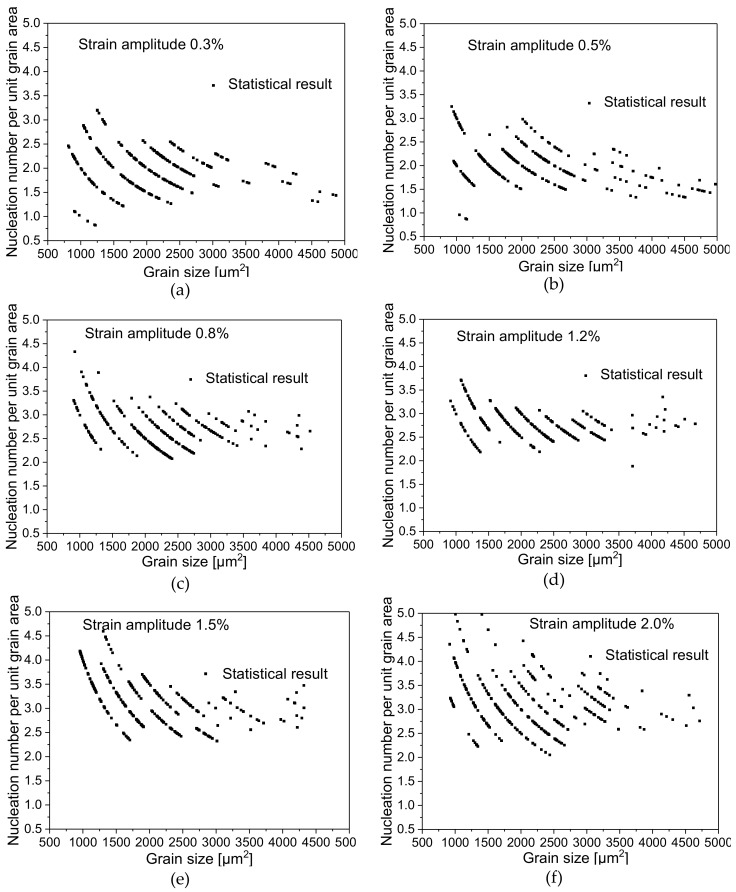
Distribution diagram of TNN under per unit grain size area at different strain amplitudes: (**a**) 0.3%; (**b**) 0.5%; (**c**) 0.8%; (**d**) 1.2%; (**e**) 1.5%; (**f**) 2.0%.

**Figure 9 materials-13-02454-f009:**
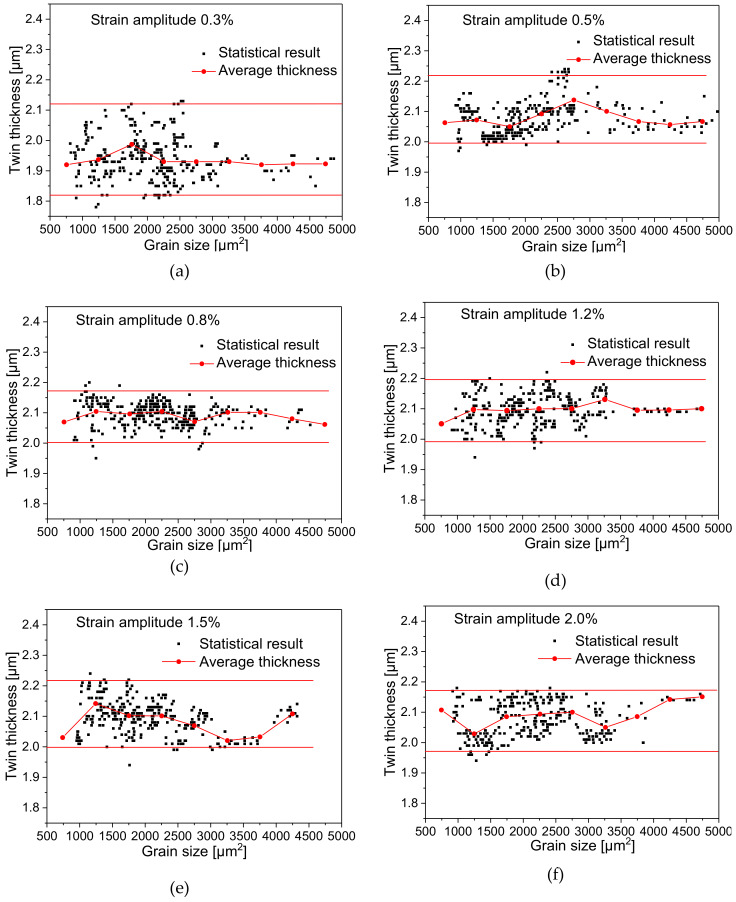
Distribution diagram of the relationship between grain area size and average twin thickness at different strain amplitudes: (**a**) 0.3%; (**b**) 0.5%; (**c**) 0.8%; (**d**) 1.2%; (**e**) 1.5%; (**f**) 2.0%.

**Figure 10 materials-13-02454-f010:**
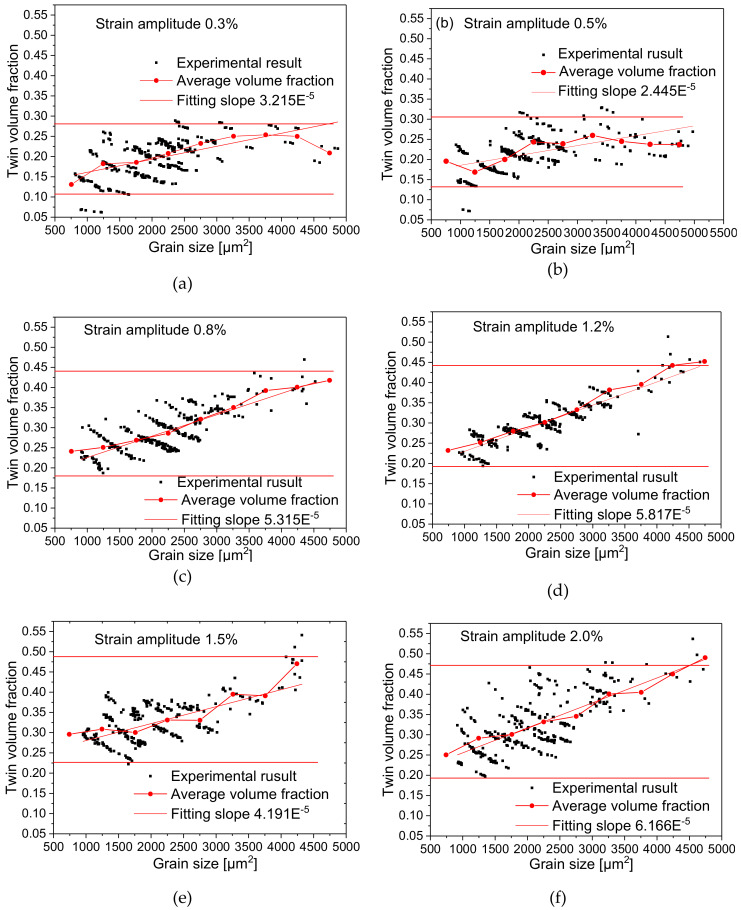
Distribution diagram of ratio of twin area to grain area at different strain amplitudes: (**a**) 0.3%; (**b**) 0.5%; (**c**) 0.8%; (**d**) 1.2%; (**e**) 1.5%; (**f**) 2.0%.

**Table 1 materials-13-02454-t001:** Chemical composition of AZ31 magnesium alloy (mass fraction/%)**.**

Al	Zn	Mn	Si	Ni	Fe	Mg
3.01	0.9	0.5	0.04	0.005	0.005	Bal.

**Table 2 materials-13-02454-t002:** The average fatigue life at different strain amplitudes.

Strain Amplitude	0.3%	0.5%	0.8%	1.2%	1.5%	2.0%
Average cyclic numbers	8601	1718	431	228	164	89

**Table 3 materials-13-02454-t003:** Distribution of the relationship between strain and average grain size.

Strain Amplitude	0.3%	0.5%	0.8%	1.2%	1.5%	2.0%
Number of grains	334	312	332	321	330	305
Average grain size [μm^2^]	2088	2179	2198	2186	2048	2144

**Table 4 materials-13-02454-t004:** Distribution of the relationship between strain amplitudes and average twin nucleation number (TNN).

Strain Amplitude	0.3%	0.5%	0.8%	1.2%	1.5%	2.0%
maximum numbers of twin nucleation(band)	8	8	13	14	15	15
average numbers of twin nucleation(band)	4.11	4.28	5.87	5.94	6.22	6.5
minimum numbers of twin nucleation(band)	1	1	3	3	4	3

**Table 5 materials-13-02454-t005:** Distribution of the relationship between strain and average twin thickness.

Strain Amplitude	0.3%	0.5%	0.8%	1.2%	1.5%	2.0%
Upper bound of twin thickness [μm]	2.1	2.21	2.17	2.19	2.22	2.17
Average twin thickness [μm]	1.94	2.08	2.09	2.1	2.09	2.08
Lower bound of twin thickness [μm]	1.81	1.98	2.01	1.99	1.98	1.96

Note: the upper and lower bound of twin thickness is an estimated value of the distribution of twin thickness, it normally represents the limit of twin thickness for most grain sizes.

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
