# Peer review of "Statistical Analysis of Grain-Scale Effects of Twinning Deformation for Magnesium Alloys under Cyclic Strain Loading"

_materials, 2020, doi:10.3390/ma13112454_

Round 1

Reviewer 1 Report

Dear authors,

The aim of the paper, i.e. description of deformation development and its statistical analysis, is interesting and important for other researchers. It was written in a clear but simplified manner. For example, in 298 line:

  • 1st sentence: “ The twin volume fraction …..”
  • 2nd : “ In …. twin volume fraction …”
  • 3rd : “The multi …. the twin volume fraction …”
  • 4th : “As …. the twin volume fraction ….”
  • 5th: “Regarding …. twin volume fraction ….”
  • 6th: “ However …. twin volume fraction ….”
  • 7th: “By comparing …. twin volume fraction ….”

That's how it is throughout the article. These are repetitions that are not a mistake but look bad in the text. A comprehensive linguistic correction is indicated here.

Other remarks:

115 line - … and 2200 Nm for torque - this phrase suggests that stretching / compression was combined with twisting, remove it because it was not used, but confused.

Figures 2 and next - Try to maintain the same scale on the axes in all drawings. If you can't, make it very clear. Symbols (a), (b) and following should be placed on the left side of the drawing, not in the box.

Fig. 3 and 4 - The marker should be larger and on a contrasting (white) background. T3, T4 symbols should be at the bottom and W3, W4 at the top of the figure.

Fig. 5 and next - "Grain size /mm2" is incorrect, it should be "Grain size [mm2]”, also in the following pictures.

227 - … average grain size … - what average grain size, define it.

236 - Write briefly: The histograms (a-f) as shown in Figure 6 …. Also in the following text (246, 279, 298).

287 - … interval of 500mm2n range - should it not be: … interval of 500 mm2 n-range ? After the number before his called, should by a 1 space break - correct throughout the text.

Reviewer 2 Report

Recommendation for the manuscript "Statistical analysis of grain-scale effects of twinnin deformation for magnesium alloys under cyclic strain loading" is accept in present form.

Reviewer 3 Report

Dear authors,

Please consider the following for further improvement of the manuscript:

Please double-check the English language and grammar. Some sentences are too long, e.g., 13-17 in the abstract.

Please describe what do you mean by „Magnesium alloy”, „ZK60” etc. at the first appearance.

There is no need to indicate the unit in Figure 1 caption, as it can be seen in the figure.

Please describe why there is an increase of lifetime between 0.8 and 1.2 % strain levels (Table 2.).

What was the initial grain texture and grain size in the specimen? Were they the same?

I would recommend showing one curve (e.g., the half-cycle) of each strain level in one graph.

Please magnify Figure 3. (highlighted areas) and Figure 4., and include a scale bar.

Please put the units into brackets ().

Please round up the grain sizes in Table 3.

What is the R2 of the Weibull fit in Figure 5.? How the R2 would change if Gaussian distribution would be fitted onto these results?

From Figure 7., it looks like there is a linear relationship between the grain sizes and the twin distribution in each case. Also, if we would fit a straight line to these data, the slope looks increasing with the increasing strain level. Please describe it to the reader. This observation also can be done in Figure 10.

Please describe how can multiple nucleation numbers be related to one grain size (Figure 8.).

From Figure 9. it looks like the variation of twin thickness significantly decreases after ~ 2500 µm grain size in each case. Please describe if this observation is valid.

Reviewer 4 Report

There is a small discrepancy in Fig. 2., where the (d) subfigure is labelled a different sample than in figure caption

Reviewer 5 Report

1) Did the authors estimate twin variants?

2) Between the value of units and the unit dimension should be spacing.
